# Electromyography-Triggered Constraint-Induced Movement Cycling Therapy for Enhancing Motor Function in Chronic Stroke Patients: A Randomized Controlled Trial

**DOI:** 10.3390/bioengineering11090860

**Published:** 2024-08-23

**Authors:** Jaemyoung Park, Kyeongjin Lee, Junghyun Kim, Changho Song

**Affiliations:** 1Department of Physical Therapy, Sahmyook University, Seoul 01795, Republic of Korea; 2Department of Physical Therapy, College of Health Science, Kyungdong University, Wonju 24764, Republic of Korea; 3Biomedical Research Institute, Seoul National University Hospital, Seoul 03080, Republic of Korea

**Keywords:** bioengineering, postural balance, stroke, electromyography, muscle strength

## Abstract

This single-blind randomized controlled trial investigated the effectiveness of surface electromyography (sEMG)-triggered constraint-induced movement cycling therapy (CIMCT) in improving balance, lower extremity strength, and activities of daily living in patients with chronic stroke. The participants included patients with chronic stroke-induced hemiplegia who had been diagnosed for more than 6 months, with a minimum score of 24 points on the Mini-Mental State Examination and above level 3 on the Brunnstrom stages. The trial lasted 4 weeks and participants were divided into a CIMCT group and a general cycling training (GCT) group. The CIMCT group (n = 20) used an sEMG-triggered constrained-induced movement therapy device, whereas the GCT group (*n* = 19) used a standard stationary bicycle. The primary outcome measures showed a significant increase in muscle strength on the affected side in the CIMCT group, as assessed by a manual muscle tester (*p* < 0.05), with a large effect size (d = 1.86), while no meaningful improvement was observed in the GCT group. Both groups demonstrated significant improvements in dynamic balance, as measured by the Timed Up and Go (TUG) test (*p* < 0.05), with the CIMCT group showing superior results compared to the GCT group, reflected by a large effect size (d = 0.96). Additionally, both groups showed significant improvements in balance as assessed by the Berg Balance Scale (BBS) and the Functional Reach Test (FRT). The CIMCT group exhibited more pronounced improvements than the GCT group, with large effect sizes of 0.83 for the BBS and 1.25 for the FRT. The secondary outcome measures revealed significant improvements in activities of daily living in both groups, as assessed by the modified Barthel index (MBI), with the CIMCT group achieving a substantial improvement (*p* < 0.05), accompanied by a large effect size (d = 0.87). This study concludes that sEMG-triggered CIMCT effectively improved muscle strength, postural balance, and activities of daily living in patients with chronic stroke.

## 1. Introduction

Stroke remains one of the leading causes of death and disability worldwide, with over 12 million incident cases and 6.55 million deaths reported globally in 2019. Despite advancements in treatment, the burden of stroke has increased significantly, particularly in low- and middle-income countries [1]. Stroke imposes a significant financial burden on healthcare systems globally. The cost of stroke care varies considerably across countries, with the United States reporting the highest average per-patient annual cost at USD 59,900, followed by Sweden at USD 52,725 and Spain at USD 41,950. The lifetime cost per patient can also be substantial. For instance, in Australia, the lifetime cost per patient has been estimated at USD 232,100 across all identified definitions of stroke [2]. Stroke often results in long-term disability and profoundly affects the lives of affected individuals [3]. Approximately 80–90% of stroke patients experience impaired mobility [4]. Although 75% of these patients achieve independent gait within 3 months post-stroke, around 25% continue to require assistance for gait [5]. The primary challenge lies in the impediment to smooth gait due to limb paralysis on the affected side. Thus, enhancing gait capability is a critical goal in stroke rehabilitation [6].

Patients with stroke typically exhibit an asymmetric gait, often adopting compensatory movement patterns to counteract this [7]. Post-stroke gait is characterized by a shortened stance phase and an elongated swing phase on the affected side. This gait abnormality is primarily due to lower limb muscle weakness and impaired balance control. Spatiotemporal gait asymmetry is closely associated with muscle weakness [8,9]. Moreover, balance impairment exacerbates these gait issues, leading to decreased gait speed, reduced stride length, and further spatiotemporal asymmetry [10,11].

Recent technological advancements, such as robot-assisted therapy [12,13], virtual reality rehabilitation [14,15], and wearable assistive devices [16,17,18], have spurred various studies aimed at enhancing gait rehabilitation in patients with stroke. These cutting-edge technologies represent novel approaches that facilitate treatment and redefine the role of therapists. When integrated into clinically effective treatments, the aforementioned technologies can yield superior outcomes. One method garnering significant attention in the upper-extremity rehabilitation of patients with stroke is constraint-induced movement therapy (CIMT) [19]. This therapy involves restricting the use of the unaffected limb, thereby necessitating the use of the affected limb [19]. Originally aimed at activating the affected limb through enforced disuse of the unaffected side, CIMT has evolved to focus on the active use of the affected limb through repetitive and detailed movement training [20]. Although primarily applied to the upper extremities, recent research has begun to explore the application of the modality to the lower extremities. These include studies that incorporate lower extremity orthoses, shoe insoles [21,22], and stationary bicycles [23]. Surface electromyography (sEMG) is a technique used to measure and analyze the electrical signals of muscle activity through electrodes placed on the skin’s surface [24,25]. By detecting and analyzing these electrical signals, which result from physiological changes in the muscle fibers, sEMG can provide valuable insights into muscle function and its role in movement [25]. A key application of EMG is in the use of electromyographic feedback to encourage specific movements; for instance, a previous study utilized EMG feedback to promote pedaling on the affected side in stroke patients, leading to improved muscle activation and gait on the paralyzed side [23]. However, there is still a significant gap in understanding how targeted interventions like sEMG-triggered pedaling specifically enhance muscle activation and promote functional recovery in the affected limbs of stroke patients.

In this study, we carefully selected and examined variables that are known to influence gait asymmetry, including muscle strength, static balance, and dynamic balance. Our objective was to explore how the use of sEMG feedback to encourage pedaling on the affected side could lead to improvements in these key areas, ultimately resulting in enhanced gait symmetry. Furthermore, we investigated whether this intervention could also improve daily living activities, offering a more comprehensive understanding of its potential benefits in stroke rehabilitation. We aimed to determine if this approach could indeed provide such advantages.

## 2. Materials and Methods

### 2.1. The Study Design

The study was structured as a single-blind randomized controlled trial spanning 4 weeks of intervention. The protocol, registered under the ClinicalTrials.gov identifier NCT06367140, was approved by the Ethics Committee of Kyungdong University (1041455-201706-HR-012-01) in accordance with the ethical standards outlined in the Declaration of Helsinki. All procedures adhered to the guidelines and regulations. Written informed consent was obtained from all participants after they were briefed about the research prior to the commencement.

### 2.2. Participants

This study included patients with stroke admitted to K Hospital in Seoul. The inclusion criteria for the participants were as follows: patients with chronic stroke-induced hemiplegia diagnosed with stroke more than 6 months prior to the start of the trial, achieving a minimum score of 24 points on the Mini-Mental State Examination [26], and demonstrating motor recovery at or above level 3 according to the Brunnstrom stages [27].

The exclusion criteria encompassed individuals with neurological damage unrelated to their stroke, orthopedic issues such as fractures or peripheral nervous system damage in the lower limbs, visual or auditory impairments, those who had experienced more than one stroke, and those with less than 80% participation rate in the study. 

Prior to the commencement of the study, all participants were informed of its purpose and procedures, and only those who voluntarily signed the consent form were included in the study. The sample size for this study was determined based on the results of a pilot study. Before conducting the main experiment, a pilot study was performed with 5 participants in each group, using the same methods as the main study. The effect size was calculated to be 0.87 based on the Timed Up and Go (TUG) test, a dynamic balance variable. G-power 3.1.9 software was employed to compare the two groups’ differences. The significance level was set at 0.05, and the power of the test was 0.8 [28], resulting in a required sample size of 18 participants per group. To account for a potential dropout rate of 10%, the final sample size was increased to 20 participants per group.

### 2.3. Experimental Procedures

We recruited 58 stroke patients receiving treatment at the K Rehabilitation Hospital in Seoul, and 40 patients who met the inclusion and exclusion criteria were selected. Before participating in the experiment, a screening test was conducted to determine whether the patients met the researchers’ eligibility criteria. A total of 18 people who did not meet the inclusion criteria were excluded. After the pre-test, 20 participants were randomly allocated to the CIMCT group and 20 to the general cycling training (GCT) groups. To minimize selection bias, a computer program (random allocation software, version 1.0) was used [29], and random allocation was performed such that sex, side of paralysis, cause of stroke, disease duration, and cognitive ability were homogeneous. This task was carried out by researchers not involved in the training and assessment. The CIMCT group performed sEMG-triggered CIMCT for 50 min per session, five times a week for 4 weeks, and the GCT group performed general cycling training for 50 min per session, five times a week for 4 weeks. The training was conducted in alignment with the inpatient treatment schedule, which included two therapy sessions per day. Accordingly, training was administered once in the morning or afternoon, depending on each participant’s schedule. Although the exact timing varied between participants, the procedures were executed with consistent time intervals between sessions to ensure uniformity across the study. As a pre-test, the participants’ general characteristics, postural balance, lower limb strength, and activities of daily living were examined.

Those who could not participate in the program because of changes in their medical condition during the study were excluded from the final study. In the GCT group, 19 people participated in the study as one patient was discharged.

Ultimately, 20 participants in the CIMCT group and 19 in the GCT group took the same post-test as the pre-test and statistical analyses were performed on those who took the post-test without excluding anyone. Figure 1 presents the study process.

### 2.4. Intervention

The sEMG-triggered CIMCT device used in this study represents a novel integration of bioengineering principles into rehabilitation. The device, comprising a myoelectric sensing unit, stationary bicycle, and control unit, uses four-channel sEMG sensors to capture muscle activity signals. The signals were processed into packets using an Arduino Pro Mini 328 (Arduino Pro Mini 328, SparkFun Electronics, Niwot, CO, USA), and then transmitted to the control unit via Bluetooth (Figure 2). These signals are transmitted via Bluetooth to a control unit that visualizes the data, providing patients with real-time feedback and enabling precise adjustments to therapy based on muscle activation. To minimize interference during the sEMG-triggered CIMCT study, we conducted the data collection in a space free from other electronic equipment. Additionally, to reduce interference caused by the sEMG cables touching each other, mesh coverings were applied to the cables.

The muscle activation signals were collected using four sEMG sensors (MyoWare Muscle Sensor, SparkFun Electronics, Niwot, CO, USA), which have a high-input impedance of 110 GΩ. The sEMG electrode patches (Ag/AgCl surface electrode H2223H, 3M, Maplewood, MN, USA) were placed on the rectus femoris, biceps femoris, tibialis anterior, and gastrocnemius muscles. Each electrode was positioned according to the SENIAM project recommendations as follows:

Rectus Femoris: The electrodes were placed at 50% of the distance along the line from the anterior superior iliac spine to the superior part of the patella.

Biceps Femoris: The electrodes were placed at 50% of the distance along the line between the ischial tuberosity and the lateral epicondyle of the tibia.

Tibialis Anterior: The electrodes were placed at 1/3 of the distance along the line from the tip of the fibula to the tip of the medial malleolus.

Gastrocnemius: The electrodes were placed on the most prominent bulge of the muscle.

Although skin impedance was not directly measured, we minimized it by cleaning the skin with alcohol wipes before electrode application. These precautions helped ensure reliable sEMG signals and minimized any potential impact on the normalization process.

After applying the electrodes, participants pedaled a stationary bicycle (Motomed Viva 2, RECK-Technik GmbH & Co, Betzenweiler, Germany) at a comfortable speed for 30 s while sEMG data were collected. The average muscle activity during this period was calculated and used for normalization. A baseline threshold was defined as the average value of the normalized muscle’s reference voluntary contraction (RVC) and was used to control the increase in bicycle speed.

When participants reached their threshold, the bicycle’s speed increased by one step, and if they did not reach the threshold within 5 s, the speed decreased by one step. The amount of speed change per step followed the settings defined on the stationary bicycle. At the start of the exercise, the initial threshold was set at 150%, but this could be adjusted based on the therapist’s judgment.

For sEMG-triggered CIMCT, the participants started with a 10-min warm-up at a comfortable speed, followed by therapist-guided acceleration and deceleration. The session comprised three 10-min periods interspersed with 1-min rests and concluded with a 5-min cool-down [30]. The therapist continuously monitored the participants for discomfort or dizziness, allowing breaks or cessation at any time.

For GCT, a regular stationary bicycle without an sEMG-triggered device was used, and the training time was the same as that for the CIMT cycling training. Moreover, the training was performed under the supervision of a guardian or caregiver and consisted of a 10-min warm-up, 30 min of main exercise, and 5 min of cool-down exercise. In the general pedaling training group, patients who experienced dizziness or complained of difficulty during the training were asked to take a break and train again. The participants were instructed to stop at any time according to their will. The evaluators were blinded to the intervention details of the participants.

### 2.5. Outcome Measurements

This study measured static balance using the GB300 postural measurement system (Metitur Ltd., Jyvaskyla, Finland). This system comprises a movable triangular platform and a scale that indicates the position of the feet. The GB300 system measures sway in the standing posture, recording medial–lateral (M–L) and anterior–posterior (A–P) movements. Sway is quantified by the velocity of movement and the area covered per second, referred to as the velocity moment. This system is widely used for assessing balance in various populations, including athletes, elderly individuals, stroke patients, and those with hemiplegia. The sampling frequency was set at 50 Hz. Participants were instructed to stand with their eyes open, facing forward, for 30 s while positioned on the device. This procedure was repeated three times. Subsequently, participants performed the same posture with their eyes closed and facing forward for an additional three measurements of 30 s each.

The Timed Up and Go (TUG) test was used to assess balance ability in this study. In the test, the participant sits in a chair with armrests, rises from the chair at the same time as the word “start” is spoken, walks at their most stable and comfortable speed to a point 3 m in front of them, and then returns to and sits down in the chair at which point the time is recorded. It has a high intra-rater reliability (r = 0.99) and inter-rater reliability (r = 0.98) [31]. The raters performed three measurements using a stopwatch and recorded the average value. The TUG test primarily assesses dynamic balance and mobility by measuring the time taken to stand up, walk a short distance, turn, and sit down.

The Berg Balance Scale (BBS) is used to assess functional balance in a wide range of participants, including older individuals at a high risk of falling and patients with acute and chronic diseases. Moreover, BBS is a functional balance test method that considers three aspects of functional balance: postural maintenance, postural control by manual exercise, and response to external perturbations. The test consists of 14 items that are common in daily activities, including getting up from a sitting position, standing without holding on, sitting without leaning back, sitting from a standing position, moving between chairs, standing with eyes closed, standing with feet together without holding on, reaching forward with arms from a standing position, picking up objects from the floor, turning left and right, spinning in place, alternating feet on a step stool, standing with one foot in front of the other, and standing on one leg; all 14 items are scored on a 5-point scale ranging from 0 to 4, with high scores indicating improved performance. The perfect score on this scale is 56, with a score below 45 indicating a risk of falling. This measure has high reliability and internal validity for assessing balance ability, with intra- and inter-rater reliabilities of r = 0.99 and r = 0.98, respectively [32]. The Berg Balance Scale (BBS) is a comprehensive measure that assesses both static and dynamic balance through a series of tasks, including sitting, standing, and transferring between positions.

The Functional Reach Test (FRT) assesses the limits of physical stability and measures dynamic balance and flexibility while the participant performs a functional task. The FRT measures the maximum distance a participant can extend their arm forward from a standing position while maintaining fixed support. The distance was measured in centimeters using a Laser Rangefinder (DLE50, BOSCH, Gerlingen, Germany). The results represent the averages of three consecutive measurements. The reliability of this test was 0.89 [33]. The Functional Reach Test (FRT) evaluates the limit of stability, specifically assessing how far an individual can reach forward without losing balance.

A manual muscle tester (Model 01163, Lafayette, LA, USA, 2003) was used to evaluate lower extremity muscle strength in this study. The knee extensors, knee flexors, dorsiflexors, and plantar flexors, which are primarily responsible for the pedaling motion, were assessed. Moreover, both sides were evaluated. Patients were seated on a fixed chair with a backrest. The knee extensors were measured with a pressure plate placed on the anterior aspect of the ankle, and the patient was instructed to extend the knee. The knee flexors were assessed with a pressure plate placed on the heel of the patient in the prone position. Ankle motion was gauged in a long sitting position with the knees extended. Dorsiflexors were measured with a pressure plate positioned on the distal aspect of the dorsum of the foot and plantar flexors were assessed on the distal aspect of the sole. The average of two measurements after each exercise was recorded.

The modified Barthel index (MBI) developed by was used to measure the perfor-263 mance of daily living behaviors. The modified Barthel index (MBI) developed by was used to measure the perfor-263 mance of daily living behaviors. The MBI consists of 10 items: self-care, bathing, feeding, climbing stairs, dressing, bowel control, bladder control, gait, and transferring. The scoring system ranged from 5 to 15, with a score of 100 if all items could be performed completely independently. The inter-rater reliability was 0.93–0.98, and the Cronbach’s alpha value was 0.84 [34].

### 2.6. Statistical Analysis

Statistical analyses were performed using SPSS version 24 (IBM Corp., Armonk, NY, USA). The data were tested for normality using the Shapiro–Wilk test, and the mean and standard deviation were calculated. After the normality test, when the normal distribution assumption was satisfied, the sociodemographic characteristics of the participants were analyzed in real numbers, percentages, means, and standard deviations. The independent sample *t*-test and chi-square test were used to test the homogeneity between groups. The changes in the dependent variables before and after the intervention were analyzed using paired *t*-tests. For instances where significant differences were observed, repeated measures ANOVA was employed to compare the effects between the two groups and analyze the interaction effects between group and time. In addition, to determine the effect before and after training, the effect size of the training was investigated by dividing the difference between before and after training by the average deviation. The effect size was classified based on the minimal detectable change (MDC) of the measurements to complement related statistical methods. The MDC was adjusted for samples from two different measurements by multiplying the standard error of measurement (SEM) by 1.96, corresponding to a 95% confidence interval, and further multiplying by the square root of 2. Furthermore, SEM was estimated as the pooled standard deviation of the pre- and post-training assessments multiplied by the square root of (1 − r), where r is the ICC. All statistical significance levels (α) of the data were set at 0.05.

## 3. Results

### 3.1. General Characteristics

The CIMCT and GCT groups were homogeneous in all general characteristics including age, height, weight, stroke type, and paralyzed side (Table 1).

### 3.2. Muscle Strength

The strength of the knee flexor, extensor, dorsiflexor, and plantar flexor muscles on the affected side significantly increased after training in the CIMCT group (*p* < 0.05), with Cohen’s d values indicating a large effect size (d > 0.80). However, no significant differences were observed in the GCT group. The muscles on the healthy side demonstrated no significant differences after training in either group (Table 2 and Table 3).

### 3.3. Static Balance

Changes in static balance ability, medial and lateral sway velocity, anteroposterior sway velocity, and moment velocity exhibited no significant differences after the intervention in either group, regardless of whether the eyes were open or closed (Table 4).

### 3.4. Dynamic Balance

For the TUG, BBS, and FRT, the CIMCT group demonstrated a significant reduction after training (*p* < 0.05), and the GCT group also displayed a significant decrease after training (*p* < 0.05). Additionally, Cohen’s d values for BBS and FRT indicated a large effect size (d > 0.80). However, when comparing the differences between the groups by training method, the CIMCT group exhibited a significantly greater improvement compared to the GCT group (*p* < 0.05) (Table 5).

### 3.5. Activities of Daily Living

For the MBI, both the CIMCT and GCT groups demonstrated a significant increase after training (*p* < 0.05), with Cohen’s d values indicating a large effect size (d > 0.80). However, when comparing the differences between the groups according to the training method, the CIMCT group displayed a significantly greater improvement than the GCT group (*p* < 0.05) (Table 6).

## 4. Discussion

The present study investigates the effects of sEMG-triggered CIMCT, which focuses on constraining the movement of the unaffected side while inducing voluntary contractions on the affected side. The study also assesses muscle strength, postural balance, and activities of daily living in patients with chronic stroke.

Prior to the current study, we conducted a pilot study to investigate the muscle activation patterns of the affected side during the pedaling exercise. Based on the results of the pilot study, we developed an sEMG-triggered pedaling device that enabled pedaling exercises with muscle signals from the affected side. Continual monitoring of the four muscles and adjustment of their speed according to their activation provided feedback on muscle actions to patients, which motivated their participation. This method actively engaged the affected side, leading to improvements in gait function, which can be considered an advanced form of CIMT. In previous studies, the use of sEMG-triggered CIMCT was shown to improve spatiotemporal gait parameters and gait symmetry [23]. Building on these findings, the current study investigated changes in variables influencing spatiotemporal gait parameters and symmetry by evaluating muscle strength and balance as primary outcome measures. This approach aimed to address the significant gap in understanding how sEMG-triggered CIMCT contributes to enhancing spatiotemporal gait parameters and symmetry. Additionally, the study examined whether improvements in these variables could ultimately positively impact activities of daily living.

In the present study, training that induced voluntary contractions on the affected side enhanced muscle strength and function. The main muscles involved in pedaling are the knee extensors, knee flexors, dorsiflexors, and plantar flexors [35]. Before training, a significant difference in muscle strength between the affected and unaffected sides was noted, with approximately twice the strength in the upper leg and three times the strength in the lower leg. The effect sizes were notably large for knee extensors (1.86), knee flexors (0.98), dorsiflexors (1.29), and plantar flexors (1.16). These results satisfied the MDC criterion for muscle strength changes on the affected side, indicating statistical significance. The CIMCT group demonstrated significant improvements on the affected side and an enhancement in symmetry ratios after the intervention.

The proposition is that sEMG-triggered pedaling training, built upon electromyographic biofeedback and constraint-induced movement therapy (CIMT) principles for the lower extremities, induces improvements in muscle strength by engaging the motor cortex. This is achieved through the focused use of the affected muscles, promoting activity in the primary sensorimotor cortex (SMC) by leveraging neuroplastic mechanisms. The training combines volitional activation of the impaired side with continuous somatosensory feedback, which facilitates the reorganization of neural circuits. This process is more effective when the pyramidal tract (PT) is intact, leading to sustained long-term improvements. Even if the PT is compromised, consistent biofeedback training can still promote recovery through compensatory activation, though the effects may be less stable [36].

Improvement in postural balance is a secondary effect of pedaling training in patients with stroke. Lee [37] reported significant improvements in stability limits after 6 weeks of pedaling training in patients with stroke. However, Kim et al. [38] demonstrated no significant difference in BBS scores in patients with chronic stroke after pedaling training; although, significant differences were observed in TUG and gait speed.

In this study, static and dynamic postural balance was also assessed. Static balance and the ability to maintain posture demonstrated significant improvements in both groups; however, no significant differences were observed between them. The intervention in this study was more effective for dynamic balance than for static balance, as it induced reciprocal movements of both feet by inducing muscle contractions in the affected lower extremities. Dynamic balance was assessed using TUG, BBS, and FRT, and both groups exhibited significant improvements, with the sEMG-triggered training group displaying greater improvements than the general pedaling training group. The TUG time decreased by 12.93%. As the TUG test involves gait, sitting, standing, and turning, the test is likely to be related to improvements in gait. The BBS, which assesses the risk of falls, improved by 14.37%; however, the scores remained below 45, suggesting that the risk of falls was not eliminated. The FRT, which evaluates the limits of stability, demonstrated an improvement of 25.73%. Shen et al. [39] conducted a systematic review and meta-analysis on the effects of pedaling training on mobility and quality of life in patients with stroke and reported significant improvements in BBS, and [40] and Tang A, et al. [41] also reported increases in BBS in groups that used electronic gait training.

The improvements in postural balance observed in this study are thought to have been influenced by the improvement in muscle strength of the affected lower extremity. This, in turn, impacts the symmetry of the lower extremities. This aligns with previous research findings, suggesting that improvements in muscle tension, endurance, joint flexibility, and symmetry of the lower extremities enhance balance capabilities [42,43].

Another secondary effect observed in this study was the change in activities of daily living. Both groups demonstrated significant improvements in MBI scores, with the sEMG-triggered training group exhibiting a significant improvement (16.53%). The results are consistent with those of previous studies, indicating that task-oriented training [44] and voluntary movements of the affected lower extremities enhance cerebral reorganization and cortical restoration [45]. The aforementioned finding also aligns with research reporting a high correlation between balance and activities of daily living [46].

This study demonstrated that sEMG-triggered CIMCT effectively enhances lower limb muscle strength, postural balance, and activities of daily living in chronic stroke patients. The biofeedback device played a crucial role in engaging patients actively and improving motor function. Compared to previous studies, our research provides additional evidence supporting the practicality and effectiveness of sEMG-triggered devices. However, this study has certain limitations. Although the sample size was calculated, considering the complex pathologies of patients with stroke, representing all stroke populations is challenging. When cycling is used as an intervention, quantitatively comparing whether all patients are trained at the same intensity and duration within a given time is difficult. Additionally, the sustainability of the exercise effects could not be ascertained because follow-up observations were not conducted after the intervention.

## 5. Conclusions

This study confirms the effectiveness of sEMG-triggered CIMCT in activating voluntary muscle contractions in the lower extremities during chronic stroke rehabilitation. The findings suggest that sEMG-triggered CIMCT is a viable method for enhancing muscle strength, postural balance, functional activities, and activities of daily living in clinical settings. This approach offers a promising option for rehabilitation practitioners aiming to improve overall functional outcomes in stroke recovery. Future research should focus on integrating bioengineering technologies into rehabilitation programs and exploring their long-term benefits across different patient groups.

## Figures and Tables

**Figure 1 bioengineering-11-00860-f001:**
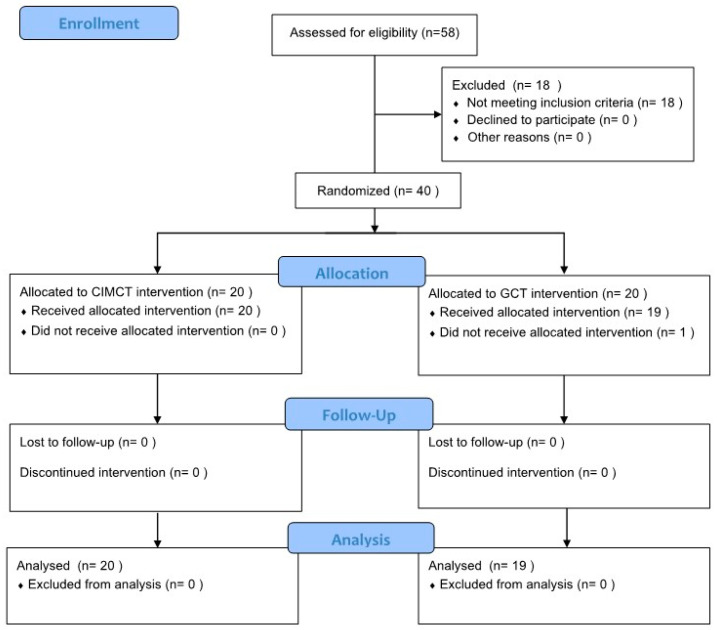
Flow chart.

**Figure 2 bioengineering-11-00860-f002:**
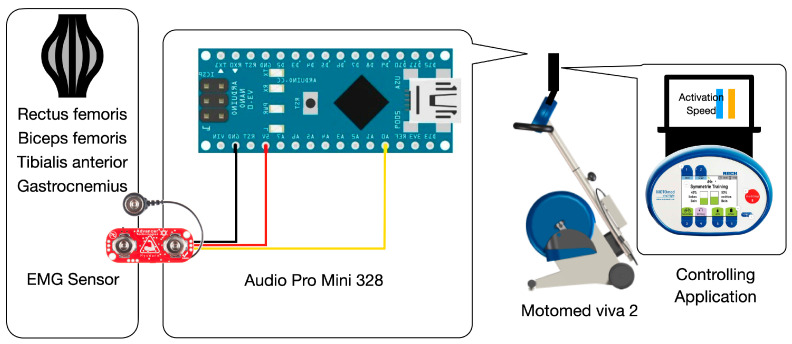
sEMG-triggered CIMCT system.

**Table 1 bioengineering-11-00860-t001:** General characteristics of the subjects.

							*n* =	39
	CIMCT			GCT			χ^2^/*t*	*p*
	*n* = 20			*n* = 19				
Age (year)	63.00	±	6.96	65.00	±	9.81	0.737	0.466
Height (cm)	162.15	±	7.74	162.58	±	8.69	0.163	0.871
Weight (kg)	58.89	±	8.32	60.17	±	8.69	0.470	0.641
BMI (point)	22.35	±	2.32	22.66	±	1.79	0.475	0.638
Duration of stroke (month)	13.85	±	6.09	17.00	±	5.83	1.648	0.108
MMSE-K	25.80	±	1.32	25.58	±	1.02	0.583	0.563
MBI	53.41	±	10.16	56.47	±	10.11	0.944	0.351
Gender (male/female)	9	/	11	10	/	9	0.634	0.227
Paretic side (right/left)	9	/	11	13	/	6	0.140	2.174
Stroke type (Infarction/hemorrhage)	13	/	7	12	/	7	0.905	0.014

Note. BMI = body mass index; MMSE-K = Mini-Mental State Examination-Korean; MBI = modified Barthel index. Values are expressed as mean ± standard deviation.

**Table 2 bioengineering-11-00860-t002:** The changes in muscle strength on the affected side.

										*n* = 39	
		CIMCT			GCT			t/F	*p*	MDC	Effect Size
		*n* = 20			*n* = 19					MDC%	
RF-A (*n*)	Pre	82.52	±	35.98	98.50	±	36.31	1.380	0.176		
	Post	92.10	±	33.76	97.29	±	34.29				
	Pre–Post	9.58	±	7.49	−1.21	±	3.19	33.643 †	0	4.64	1.86
	t	5.724 *			1.658					48.43	
	*p*	0			0.116						
BF-A (*n*)	Pre	50.03	±	19.15	53.14	±	21.41	0.635	3.449		
	Post	55.86	±	21.37	53.40	±	22.95				
	Pre–Post	5.83	±	6.44	0.27	±	2.88	11.898 †	0.001	3.99	1.1
	t	4.047 *			0.404					68.50	
	*p*	0.001			0.691						
TA-A (*n*)	Pre	59.17	±	25.27	54.78	±	21.11	0.588	0.56		
	Post	64.84	±	22.22	54.80	±	20.98				
	Pre–Post	5.67	±	5.06	0.02	±	3.54	16.180 †	0	3.13	1.29
	t	5.015 *			0.023					55.27	
	*p*	0			0.982						
Gastrocne mius-A (*n*)	Pre	92.84	±	23.27	93.36	±	21.26	0.073	0.942		
	Post	98.52	±	23.92	93.87	±	19.02				
	Pre–Post	5.67	±	3.59	0.51	±	4.25	16.891 †	0	2.23	1.32
	t	7.065 *			0.519					39.23	
	*p*	0			0.61						

Note. RF = rectus femoris muscle; BF = biceps femoris muscle; TA = tibialis anterior muscle; A = affected side; and MDC = minimal detectable change. Values are expressed as mean ± standard deviation (SD). * *p* < 0.05. † indicates that the *p*-value from the repeated measures ANOVA is less than 0.05.

**Table 3 bioengineering-11-00860-t003:** The changes in muscle strength on the non-affected side.

								*n* = 39	
		CIMCT			GCT			t	*p*
		*n* = 20			*n* = 19				
RF-NA (*n*)	Pre	159.42	±	42.33	156.58	±	34.68	0.229	0.82
	Post	161.68	±	44.49	156.20	±	35.24		
	Pre–Post	2.26	±	5.37	−0.38	±	5.50	1.521	0.137
	t	1.884			0.305				
	*p*	0.075			0.764				
BF-NA (*n*)	Pre	102.07	±	32.07	102.83	±	28.73	0.078	0.938
	Post	103.32	±	32.12	103.07	±	26.44		
	Pre–Post	1.25	±	3.90	0.24	±	5.56	0.655	0.516
	t	1.431			0.192				
	*p*	0.169			0.85				
TA-NA (*n*)	Pre	151.10	±	38.12	150.88	±	33.62	0.019	0.985
	Post	153.41	±	32.25	151.89	±	32.26		
	Pre–Post	2.31	±	7.79	1.01	±	4.77	0.625	0.536
	t	1.327			0.927				
	*p*	0.2			0.367				
Gastrocne mius-NA (*n*)	Pre	177.34	±	30.15	173.17	±	29.01	0.439	0.663
	Post	176.82	±	28.50	174.98	±	27.76		
	Pre–Post	−0.52	±	7.29	1.81	±	8.20	0.938	0.354
	t	0.321			0.961				
	*p*	0.752			0.35				

Note. RF = rectus femoris muscle; BF = biceps femoris muscle; TA = tibialis anterior muscle; and NA = non-affected side. Values are expressed as mean ± standard deviation (SD).

**Table 4 bioengineering-11-00860-t004:** The changes in static balance.

								*n* = 39		
			CIMCT			GCT			t	*p*
			*n* = 20			*n* = 19				
EC	M-L speed (mm/s)	Pre	5.63	±	2.19	5.41	±	2.13	0.315	0.755
		Post	5.25	±	3.14	5.66	±	2.64		
		Pre–Post	0.38	±	3.11	−0.25	±	4.16	0.539	0.593
		t	0.546			0.264				
		*p*	0.591			0.795				
	A-P speed (mm/s)	Pre	7.49	±	2.87	7.23	±	3.67	0.249	0.805
		Post	6.88	±	2.57	7.49	±	2.80		
		Pre–Post	0.61	±	3.12	−0.26	±	3.83	0.775	0.443
		t	0.867			0.296				
		*p*	0.397			0.771				
	Velocity moment (mm^2^/s)	Pre	6.50	±	2.38	6.12	±	2.55	0.489	0.628
		Post	6.29	±	2.06	5.95	±	1.69		
		Pre–Post	0.22	±	2.61	0.17	±	2.19	0.061	0.952
		t	0.369			0.336				
		*p*	0.716			0.741				
EO	M-L speed (mm/s)	Pre	4.16	±	1.41	4.51	±	1.20	0.845	0.404
		Post	4.32	±	1.09	4.32	±	1.12		
		Pre–Post	−0.16	±	1.19	0.19	±	1.58	0.781	0.440
		t	0.610			0.515				
		*p*	0.549			0.613				
	A-P speed (mm/s)	Pre	5.75	±	1.78	5.68	±	0.91	0.160	0.874
		Post	5.68	±	0.91	5.39	±	1.53		
		Pre–Post	0.30	±	1.80	0.29	±	1.69	0.017	0.987
		t	0.745			0.753				
		*p*	0.465			0.462				
	Velocity moment (mm^2^/s)	Pre	4.89	±	2.16	4.77	±	2.14	0.174	0.863
		Post	4.34	±	1.11	4.32	±	1.12		
		Pre–Post	−1.27	±	0.90	−0.66	±	1.11	1.906	0.064
		t	1.131			0.660				
		*p*	0.272			0.518				

Note. M-L: Medial-Lateral; A-P: Anterior-Posterior; EO: Eyes Open; EC: Eyes Closed.

**Table 5 bioengineering-11-00860-t005:** The changes in dynamic balance.

										*n* = 39	
		CIMCT			GCT			t/F	*p*	MDC	Effect Size
		*n* = 20				*n* = 19				MDC%	
TUG (sec)	Pre	36.70	±	4.89	35.50	±	5.25	0.741	0.463		
	Post	31.82	±	4.56	33.70	±	4.51				
	Pre–Post	−4.88	±	3.06	−1.79	±	3.38	8.926 †	0.005	1.89	−0.96
	t	7.132 *			2.322 *					38.86	
	*p*	0			0.033						
BBS (point)	Pre	30.56	±	7.88	29.00	±	9.51	0.561	0.578		
	Post	35.05	±	7.27	30.97	±	9.12				
	Pre–Post	4.49	±	2.94	1.97	±	3.10	6.767 †	0.013	1.82	0.83
	t	6.832 *			2.793 *					40.57	
	*p*	0			0.012						
FRT (cm)	Pre	14.45	±	4.33	13.17	±	4.08	0.945	0.351		
	Post	17.83	±	4.30	13.79	±	4.59				
	Pre–Post	3.38	±	2.85	0.62	±	1.20	15.299 †	0	1.76	1.25
	t	5.315 *			2.254 *					52.15	
	*p*	0			0.038						

Note. TUG = Timed Up and Go; BBS = Berg Balance Scale; FRT = Functional Reach Test. Values are expressed as mean ± standard deviation (SD). * *p* < 0.05. † indicates that the *p*-value from the repeated measures ANOVA is less than 0.05.

**Table 6 bioengineering-11-00860-t006:** The changes in activities of daily living.

										*n* = 39	
		CIMCT			GCT			t/F	*p*	MDC	Effect Size
		*n* = 20				*n* = 19				MDC%	
MBI (score)	Pre	53.41	±	10.16	56.47	±	10.11	0.944	0.351		
	Post	62.40	±	14.77	58.34	±	9.69				
	Pre–Post	8.99	±	6.46	1.87	±	9.69	19.831 †	0	4.01	0.87
	t	6.224 *			3.058 *					44.53	
	*p*	0			0.007						

Note. MBI = modified Barthel index. Values are expressed as mean ± standard deviation (SD). * *p* < 0.05. † indicates that the *p*-value from the repeated measures ANOVA is less than 0.05.

## Data Availability

The original contributions presented in the study are included in the article; further inquiries can be directed to the corresponding author/s. https://osf.io/zmey3/, accessed on 21 August 2024 (DOI: 10.17605/OSF.IO/ZMEY3).

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
