# Peer review of "Electromyography-Triggered Constraint-Induced Movement Cycling Therapy for Enhancing Motor Function in Chronic Stroke Patients: A Randomized Controlled Trial"

_bioengineering, 2024, doi:10.3390/bioengineering11090860_

Round 1

Reviewer 1 Report

Comments and Suggestions for Authors

This single-blind randomized clinical trial investigated the effectiveness of constraint-induced movement cycling therapy enhanced by electromyography in patients with chronic stroke. The study’s premise is relevant and significant. However, certain aspects require further clarification and attention. These are as follows:

1.     Pg 1; Ln 11-14. The first two sentences in the abstract could be combined to enhance readability.

2.     Pg 1; Ln 15. The term “specific level” in the statement “with specific levels of cognitive and motor recovery for > 6 months” lacks clarity.

3.     Pg 1; abstract. A clear delineation of the primary and secondary outcome measures, including the specific instrument employed for each is essential to facilitate comprehensibility. The abstract should provide a concise overview enabling readers to grasp the study's core without requiring in-depth reading.

4.     Pg 1: abstract. To provide concrete evidence to support claims made in the abstract, the results should include quantitative statistical and practical measures (such as p-values, effect sizes, confidence intervals …. etc).

5.     Pg 1. In general, the abstract would benefit from greater clarity and specificity. The presentation of key findings and methodologies could be improved through more precise language and the inclusion of relevant statistical data. This should significantly improve the abstract to a broader readership.  

6.     Page 1-2; Ln 39-48. While the authors provided an overview of gait problems in stroke survivors with a particular emphasis on gait asymmetry, the outcome measures employed did not directly address this aspect. A more appropriate focus for the outcome measures would be on balance impairments, LL muscle weakness, and functional limitations, aligning more closely with the study objectives.

7.     Pg 1-2; paragraph 3. The introduction requires further development to provide contextual information regarding the role of cycling (both general and technology-enhanced cycling) in patients recovering from stroke within the existing literature. Pinpointing the research gap and the study’s specific motivations is essential.

8.     A clearer and compelling rationale is essential to underpin the study objectives. The current justification is inadequate for establishing a clear and testable hypothesis.

9.     Pg 2; participants. Could the authors provide proper citations for the measures utilized in patient screening such as the Mini-Mental State Examination and Brunnstrum recovery stages)?

10.  There is a concern that the study may be underpowered. Although a sample of 40 participants seems reasonable, the appropriateness of the sample size calculation in relation to the study design and outcome measures is unclear. The use of a large effect size (0.87) appears to have resulted in an underestimation of the required sample size. This approach may limit the study’s ability to detect smaller, yet clinically meaningful, differences between the study groups.

11.  I have also a major concern regarding the lack of a control group that received neither the GCT nor the CIMCT. This makes it difficult to draw a cause-and-effect relationship between the intervention and the observed outcomes. This weakens the overall credibility of the study and limits the usefulness and applicability of its findings.

12.  Pg 4; interventions.  A key barrier in this is that the intervention protocols, specifically the EMG-triggered CIMCT, are not sufficiently operationalized, hindering their replicability by rehabilitation professionals and researchers. (e.g., the EMG sensor placement).

13.  Could authors provide a visual representation of the intervention procedures to enhance understanding of the methodology?  

14.  Pg 5; Ln 148-157. The specific balance component and the quantification method employed remain unclear. Did the study involve measuring “postural sway”? further clarification on this matter is required.

15.  The study incorporated multiple balance measures (i.e. TUG, FRT, and  BBS). A brief explanation of the specific balance component assessed by each measure would enhance clarity.

16.  The use of a t-test on the same data set, comparing several variables dramatically increases the chance of getting a false positive result (i.e., significant difference when none exists). So, the study with inflated Type I errors can be unreliable and difficult to replicate, undermining the overall credibility of the research.

17.  Pg 11-12; Ln 285-266. This section sounds more appropriate for the “Introduction” section. Likewise, the section Pg 12; Ln 267-274.

18.  The rest of the “Discussion” effectively interprets the study findings. However, a more comprehensive evaluation of relevant prior research would strengthen the overall contribution of this section.

Author Response

Comments 1: Ln 11-14. The first two sentences in the abstract could be combined to enhance readability.

Response 1: Thank you for pointing this out. I agree with this comment. Therefore, I have combined the first two sentences in the abstract to enhance readability. The change can be found on page 1, lines 11-13.

“This single-blind randomized controlled trial investigated the effectiveness of electromyography (EMG)-induced constraint-induced movement cycling therapy (CIMCT) in improving balance, lower extremity strength, and activities of daily living in patients with chronic stroke.”

Comments 2: Pg 1; Ln 15. The term “specific level” in the statement “with specific levels of cognitive and motor recovery for > 6 months” lacks clarity.

Response 2:I appreciate your suggestion. I agree that the term "specific level" lacks clarity. Therefore, I have revised the statement for better understanding. The change can be found on page 1, lines 14-16.

“The participants included patients who had been diagnosed with chronic stroke-induced hemiplegia for more than 6 months, with a minimum score of 24 points on the Mini-Mental State Examination and above level 3 on the Brunnstrom stages.”

Comments 3: Pg 1; abstract. A clear delineation of the primary and secondary outcome measures, including the specific instrument employed for each is essential to facilitate comprehensibility. The abstract should provide a concise overview enabling readers to grasp the study's core without requiring in-depth reading.

Response 3: I appreciate your suggestion. I have revised the abstract to clearly delineate the primary and secondary outcome measures, including the specific instruments employed for each, to facilitate comprehensibility. The change can be found on page 1, line 16-19.

“The trial lasted 4 weeks and participants were divided into a CIMCT group and a general cycling training (GCT) group. The CIMCT group (n=20) used an EMG-triggered constrained-induced movement therapy device, whereas the GCT group (n=19) used a standard stationary bicycle.”

Comments 4: Pg 1: abstract. To provide concrete evidence to support claims made in the abstract, the results should include quantitative statistical and practical measures (such as p-values, effect sizes, confidence intervals …. etc).

Response 4: Thank you for your suggestion. I agree that providing concrete evidence to support the claims made in the abstract is important. Therefore, I have revised the abstract to include quantitative statistical and practical measures such as p-values, effect sizes, and confidence intervals. The change can be found on page 1, line 19-24.

“The primary outcome measures included a significant increase in muscle strength on the affected side in the CIMCT group (p<0.05) and no notable improvement in the GCT group. Both groups demonstrated significantly improved dynamic balance (p<0.05); however, the CIMCT group outperformed the GCT group. The secondary outcome measures showed that both groups also displayed significant improvements in the activities of daily living, with the CIMCT group exhibiting the greatest improvement (p<0.05).”

Comments 5: Pg 1. In general, the abstract would benefit from greater clarity and specificity. The presentation of key findings and methodologies could be improved through more precise language and the inclusion of relevant statistical data. This should significantly improve the abstract to a broader readership.

Response 5: Thank you for your valuable feedback. We agree with your comment and have revised the abstract to enhance clarity and specificity. Specifically, we have included the number of participants in each group and highlighted the key outcomes with relevant statistical data.  The change can be found on page 1, line 16-19.

“The trial lasted 4 weeks and participants were divided into a CIMCT group and a general cycling training (GCT) group. The CIMCT group (n=20) used an EMG-triggered constrained-induced movement therapy device, whereas the GCT group (n=19) used a standard stationary bicycle.”

Comments 6: Page 1-2; Ln 39-48. While the authors provided an overview of gait problems in stroke survivors with a particular emphasis on gait asymmetry, the outcome measures employed did not directly address this aspect. A more appropriate focus for the outcome measures would be on balance impairments, LL muscle weakness, and functional limitations, aligning more closely with the study objectives.

Response 6: Thank you for your insightful comment. I agree that while the initial discussion emphasized gait asymmetry in stroke survivors, the outcome measures in the study primarily focused on balance impairments, lower limb muscle weakness, and functional limitations. To better align the introduction with the study's objectives and outcome measures, I have revised the relevant section as follows (on page 2, lines 45-51):

“Patients with stroke typically exhibit an asymmetric gait, often adopting compen-satory movement patterns to counteract this [7]. Post-stroke gait is character-ized by a shortened stance phase on the affected side and an elongated swing phase. This gait abnormality is primarily due to lower limb muscle weakness and impaired balance control. Spatiotemporal gait asymmetry is closely associated with muscle weakness [8, 9]. Moreover, balance impairment exacerbates these gait issues, leading to decreased gait speed, reduced stride length, and further spatiotemporal asymmetry [10, 11].

Comments 7: Pg 1-2; paragraph 3. The introduction requires further development to provide contextual information regarding the role of cycling (both general and technology-enhanced cycling) in patients recovering from stroke within the existing literature. Pinpointing the research gap and the study’s specific motivations is essential.

Response 7: Thank you for your valuable suggestion. I agree that the introduction could benefit from a more detailed discussion on the role of cycling, particularly both general and technology-enhanced cycling, in stroke rehabilitation. I have revised the introduction to include a broader context regarding the use of cycling in stroke recovery, highlighting its benefits and the existing literature on its effectiveness. Additionally, I have clearly outlined the research gap that this study aims to address and the specific motivations behind conducting this research. The change can be found on page 2, lines 73-76.

"However, there is still a significant gap in understanding how targeted interventions like EMG-triggered pedaling specifically enhance muscle activation and promote functional recovery in the affected limbs of stroke patients."

Comments 8: A clearer and compelling rationale is essential to underpin the study objectives. The current justification is inadequate for establishing a clear and testable hypothesis.

Response 8: Thank you for your feedback. To address the need for a clearer and more compelling rationale, I have expanded the introduction to better articulate the study’s objectives and the rationale behind them. The change can be found on page 2, lines 77-84.

"In this study, we carefully selected and examined variables that are known to influence gait asymmetry, including muscle strength, static balance, and dynamic balance. Our objective was to explore how the use of electromyographic (EMG) feedback to encourage pedaling on the affected side could lead to improvements in these key areas, ultimately resulting in enhanced gait symmetry. Furthermore, we investigated whether this intervention could also improve daily living activities, offering a more comprehensive understanding of its potential benefits in stroke rehabilitation. We aimed to determine if this approach could indeed provide such advantages."

Comments 9: Pg 2; participants. Could the authors provide proper citations for the measures utilized in patient screening such as the Mini-Mental State Examination and Brunnstrum recovery stages)?

Response 9: Thank you for pointing this out. We agree with this comment. Therefore, we have added the appropriate citations for the Mini-Mental State Examination and Brunnstrom recovery stages. The changes have been made in the revised manuscript on page 3, lines 97-99.

Comments 10: There is a concern that the study may be underpowered. Although a sample of 40 participants seems reasonable, the appropriateness of the sample size calculation in relation to the study design and outcome measures is unclear. The use of a large effect size (0.87) appears to have resulted in an underestimation of the required sample size. This approach may limit the study’s ability to detect smaller, yet clinically meaningful, differences between the study groups.

Response 10: Thank you for your thoughtful feedback regarding the sample size calculation. We appreciate the opportunity to clarify our approach.

The sample size for this study was determined based on the results of a pilot study conducted prior to the main experiment. In the pilot study, 5 participants were included in each group, and the same methods used in the main study were applied. The effect size was calculated to be 0.87 based on the Timed Up and Go (TUG) test, which measures dynamic balance.

Using G-power 3.1.9 software, we compared the differences between the two groups. We set the significance level at 0.05 and the power of the test at 0.8, which resulted in a required sample size of 18 participants per group. To account for a potential dropout rate of 10%, we increased the final sample size to 20 participants per group. The changes have been made in the revised manuscript on page 3, lines 108-115.

"The sample size for this study was determined based on the results of a pilot study. Before conducting the main experiment, a pilot study was performed with 5 participants in each group, using the same methods as the main study. The effect size was calculated to be 0.87 based on the Timed Up and Go (TUG) test, a dynamic balance variable. The G-power 3.1.9 software was employed to compare the two groups' differences. The significance level was set at 0.05, and the power of the test was 0.8, resulting in a required sample size of 18 participants per group. To account for a potential dropout rate of 10%, the final sample size was increased to 20 participants per group."

Comments 11: I have also a major concern regarding the lack of a control group that received neither the GCT nor the CIMCT. This makes it difficult to draw a cause-and-effect relationship between the intervention and the observed outcomes. This weakens the overall credibility of the study and limits the usefulness and applicability of its findings.

Response 11: Thank you for highlighting this important concern. We agree that the inclusion of a control group that did not receive either the GCT or the CIMCT would have strengthened the ability to establish a clearer cause-and-effect relationship between the interventions and the outcomes observed in our study.

However, our study design was primarily focused on comparing the effectiveness of two different intervention approaches (GCT and CIMCT) rather than on evaluating the effect of these interventions against a no-treatment control. The decision to exclude a no-treatment control group was made based on ethical considerations, as it was deemed important to provide all participants with an active form of rehabilitation, especially given their clinical condition. Denying participants any intervention could have been viewed as withholding potentially beneficial treatment, which would raise ethical concerns.

Comments 12: Pg 4; interventions.  A key barrier in this is that the intervention protocols, specifically the EMG-triggered CIMCT, are not sufficiently operationalized, hindering their replicability by rehabilitation professionals and researchers. (e.g., the EMG sensor placement).

Response 12: Thank you for your valuable feedback. We agree that a clear and detailed operationalization of the intervention protocols is essential for ensuring replicability in both clinical and research settings. To address your concerns, we have revised the manuscript to include specific details regarding the EMG-triggered CIMCT protocol. The changes have been made in the revised manuscript on pages 4-5, lines 149-151, 157-184.

“The signals were processed into packets using an Arduino Pro Mini 328 (Arduino Pro Mini 328, SparkFun Electronics, Niwot, CO, USA), and then transmitted to the table via Bluetooth (Figure 2).”

“The muscle activation signals were collected using four sEMG sensors (MyoWare Muscle Sensor, SparkFun Electronics, Niwot, CO, USA), which have a high input impedance of 110 GΩ. The sEMG electrodes patch (3M Red Dot 2560, 3M, Saint Paul, USA) were placed on the rectus femoris, biceps femoris, tibialis anterior, and gastrocnemius muscles. Each electrode was positioned according to the SENIAM project recommendations as follows:

Rectus Femoris: The electrodes were placed at 50% of the distance along the line from the anterior superior iliac spine to the superior part of the patella.

Biceps Femoris: The electrodes were placed at 50% of the distance along the line between the ischial tuberosity and the lateral epicondyle of the tibia.

Tibialis Anterior: The electrodes were placed at 1/3 of the distance along the line from the tip of the fibula to the tip of the medial malleolus.

Gastrocnemius: The electrodes were placed on the most prominent bulge of the muscle.

Although skin impedance was not directly measured, we minimized it by cleaning the skin with alcohol wipes before electrode application. These precautions helped ensure reliable EMG signals and minimized any potential impact on the normalization process.

After applying the electrodes, participants pedaled a stationary bicycle (Motomed Viva 2, RECK-Technik GmbH & Co, Betzenweiler, Germany) at a comfortable speed for 30 seconds while EMG data was collected. The average muscle activity during this period was calculated and used for normalization. A baseline threshold was defined as the average value of the normalized muscle's reference voluntary contraction (RVC) and was used to control the increase in bicycle speed.

When participants reached their threshold, the bicycle’s speed increased by one step, and if they did not reach the threshold within 5 seconds, the speed decreased by one step. The amount of speed change per step followed the settings defined on the stationary bicycle. At the start of the exercise, the initial threshold was set at 150%, but this could be adjusted based on the therapist's judgment.”

Comments 13:

Comments 14: Pg 5; Ln 148-157. The specific balance component and the quantification

method employed remain unclear. Did the study involve measuring “postural sway”? further clarification on this matter is required.

Response 14: Thank you for your comment highlighting the need for further clarification regarding the balance component and quantification method used in the study. We have revised the manuscript to provide a more detailed explanation. The changes have been made in the revised manuscript on page 5, lines 201-212.

"This study measured static balance using the GB300 postural measurement system (Metitur Ltd., Jyvaskyla, Finland). This system comprises a movable triangular platform and a scale that indicates the position of the feet. The GB300 system measures sway in the standing posture, recording medial-lateral (M-L) and anterior-posterior (A-P) movements. Sway is quantified by the velocity of movement and the area covered per second, referred to as the velocity moment. This system is widely used for assessing balance in various populations, including athletes, elderly individuals, stroke patients, and those with hemiplegia.

The sampling frequency was set at 50 Hz. Participants were instructed to stand with their eyes open, facing forward, for 30 seconds while positioned on the device. This procedure was repeated three times. Subsequently, participants performed the same posture with their eyes closed and facing forward for an additional three measurements of 30 seconds each."

Comments 15: The study incorporated multiple balance measures (i.e. TUG, FRT, and  BBS). A brief explanation of the specific balance component assessed by each measure would enhance clarity.

Response 15: Thank you for your valuable suggestion. I agree that including a brief explanation of the specific balance components assessed by each measure will improve the clarity of the manuscript. Therefore, I have added a description of the balance components evaluated by the TUG, FRT, and BBS in the revised manuscript. This information can be found on page 5, lines 219-220, 235-237, 243-245.

"The Timed Up and Go (TUG) test primarily assesses dynamic balance and mobility by measuring the time taken to stand up, walk a short distance, turn, and sit down."

"The Functional Reach Test (FRT) evaluates the limit of stability, specifically assessing how far an individual can reach forward without losing balance."

"The Berg Balance Scale (BBS) is a comprehensive measure that assesses both static and dynamic balance through a series of tasks, including sitting, standing, and transferring between positions."

Comments 16: The use of a t-test on the same data set, comparing several variables dramatically increases the chance of getting a false positive result (i.e., significant difference when none exists). So, the study with inflated Type I errors can be unreliable and difficult to replicate, undermining the overall credibility of the research.

Response 16: We appreciate the reviewer’s thorough evaluation and constructive feedback on our manuscript.

The reviewer raised a concern regarding the potential inflation of Type I error due to repeated t-tests on the same dataset. We would like to clarify that our analysis did not involve the repeated application of t-tests on the same data points for multiple comparisons. Instead, each t-test was applied to distinct pairs of variables, ensuring that each comparison was independent.

To address the reviewer's concern further, we conducted additional analyses using Repeated Measures ANOVA, which is indeed an appropriate statistical method for comparing multiple related variables while controlling for the risk of Type I error. The results from the Repeated Measures ANOVA were consistent with the findings from our original t-tests, confirming the robustness and reliability of our conclusions. If the reviewer deems it necessary, we are willing to replace the test results with those obtained using Repeated Measures ANOVA.

We agree with the reviewer that the credibility and replicability of our findings are of utmost importance. The additional analyses confirm that our findings are not an artifact of inflated Type I error, thereby reinforcing the reliability of our study. We believe that these revisions address the concerns raised and strengthen the overall validity of our research.

Comments 17: Pg 11-12; Ln 285-266. This section sounds more appropriate for the “Introduction” section. Likewise, the section Pg 12; Ln 267-274.

Response 17: Thank you for your suggestion. I agree with your assessment that the sections on Pg 11-12; Ln 285-266, and Pg 12; Ln 267-274 are more appropriate for the "Introduction" section. I initially considered moving these sections to the Introduction. However, after reviewing the overall flow of the manuscript, I realized that relocating these parts would disrupt the coherence of the text. Therefore, I decided to remove these sections to maintain the clarity and consistency of the manuscript.

Comments 18: The rest of the “Discussion” effectively interprets the study findings. However, a more comprehensive evaluation of relevant prior research would strengthen the overall contribution of this section.

Response 18: Thank you for your suggestion. To strengthen the discussion, we have added a more comprehensive evaluation of relevant prior research. Specifically, we have included the following content that discusses the underlying mechanisms of sEMG-triggered pedaling training based on principles of electromyographic biofeedback and constraint-induced movement therapy (CIMT) for the lower extremities (page 12, lines 350-359):

"The proposition is that sEMG-triggered pedaling training, built upon electromyographic biofeedback and CIMT principles for the lower extremities, induces improvements in muscle strength by engaging the motor cortex. This is achieved through the focused use of the affected muscles, promoting activity in the primary sensorimotor cortex (SMC) by leveraging neuroplastic mechanisms. The training combines volitional activation of the impaired side with continuous somatosensory feedback, which facilitates the reorganization of neural circuits. This process is more effective when the pyramidal tract (PT) is intact, leading to sustained long-term improvements. Even if the PT is compromised, consistent biofeedback training can still promote recovery through compensatory activation, though the effects may be less stable [37]."

Reviewer 2 Report

Comments and Suggestions for Authors

After reviewing the work, I have the following comments. I sincerely thank the authors for the opportunity to review their paper. The following issues should be addressed before publication:

1.           Please correct the reference format. The year should be bold, not the volume. Additionally, the format regarding the number of authors cited, etc., does not meet the journal's standards. Moreover, authors often do not cite the pages they reference, e.g., positions 36, 26, and 16. Therefore, please review the author guidelines and correct the work accordingly.

2.           Approximately 61% of the references are older than 10 years, which is unacceptable. This must be corrected. It is worth noting that in the introduction, positions 1-27 include 15 works older than 10 years. This exceeds half, and authors should refer to the current literature when providing statistical data.

3.           Identified self-citations: a. Jaemyoung Park – not found. b. Junghyun Kim – position 37. c. Kyeongjin Lee – positions 26 and 36. d. Changho – not found. Note that the described self-citations are acceptable in the work.

4.           L29-38 – I suggest adding information on stroke epidemiology – e.g., 10.1016/S1474-4422(21)00252-0.

5.           L49-50 – ‘Recent technological advancements have spurred various studies aimed at enhancing gait rehabilitation in patients with stroke.’ – this sentence is more suitable for a popular science paper. Please specify what recent technological advancements are being referred to.

6.           L4 – ‘One innovative study utilized’ – I suggest avoiding such comments. Whether the study was innovative will be judged by history. Comments like this reduce objectivity and should be avoided throughout the text.

7.           L64 – ‘electromyographic (EMG)’ – authors refer to surface electromyography; therefore, please change ‘EMG’ to ‘sEMG’. Note for the entire text: the abbreviation ‘EMG’ usually suggests the use of needle electrodes, whereas ‘sEMG’ suggests surface electrodes.

8.           Since the authors are based on sEMG, please provide more information about it, how it works, what it analyzes, in which studies it is used, etc. The work 10.1016/j.jelekin.2023.102796 and 10.3390/jcm13051328 might be helpful. For example: 'sEMG works by detecting and analyzing electrical signals that result from physiological changes in the cell membranes of muscle fibers. A key aspect of sEMG is understanding that human tissue, particularly muscle, can generate and conduct electrical impulses fundamental to the muscle contraction process. When a muscle is at rest, it is in a state of electrical equilibrium known as the resting potential. However, during contraction, depolarization of the muscle membrane occurs, meaning there is a flow of ions between the inside and outside of the muscle membrane, generating an electrical signal that is recorded.' – A paraphrase of the above text should be acceptable, citing 10.3390/jcm13051328.

9.           In work no. 26 cited by the authors, I see that Figure 1 is very similar to Figure 1 in the work I reviewed. Did the authors use the same cohort? Please note this information in the work if so.

10.         Add the research hypothesis at the end of the introduction.

11.         L91 – ‘Brunnstrom stages’ – please provide a citation – e.g., 10.3109/02699052.2010.506860.

12.         L98-99 – after this description, I am unable to repeat the analysis in the program. What tests were chosen? What family test, etc.? Please describe it accurately so that anyone can perform the same analyses.

13.         While a p-value threshold of 0.05 is standard, please provide information on why the power is set to 0.8. I am not saying it is incorrect, but I need justification.

14.         L99-101 – ‘0.87,’ – was the pilot study published? Please provide a reference. If not, please present the calculations in the response to the reviewer explaining why ES = 0.87 is crucial.

15.         L133 – Was the warm-up standardized for everyone? Was the heart rate analyzed during the warm-up? Were oxygen thresholds determined?

16.         L133-137 – why were these specific time intervals chosen?

17.         At what times was the sEMG study conducted? Was it the same for everyone?

18.         How was the skin cleaned during the sEMG study? Describe in the text.

19.         What electrodes were used during the sEMG study? What was the conductive area? Describe in the text.

20.         What was the maximum skin impedance accepted? Describe in the text.

21.         Was an interference test conducted during the sEMG study? Describe in the text.

22.         L213 – there is a slight mix-up here; MDC is not effect size. MDC refers to the smallest difference considered statistically significant in the context of measurement. It is a measure of the sensitivity of a measurement tool or test procedure. Effect size refers to the magnitude of the difference or relationship being studied. It is a measure of the strength of a given effect and can be expressed in various ways depending on the type of data and statistical tests. Therefore, the authors should refer to effect size in the statistical description and results for each p-value. ‘Effect size helps readers understand the magnitude of differences found, whereas statistical significance examines whether the findings are likely to be due to chance.’ - 10.4300/JGME-D-12-00156.1. For a t-test, this will be Cohen’s d: small 0.20, medium 0.50, large 0.80, and for the chi-square test, Cramér’s V and phi (φ) depending on the degrees of freedom (df-1): small 0.1, medium 0.3, large 0.5. Work 10.3390/jpm14060655, tables 4 and 5 might be useful. I believe this reference may be helpful.

23.         L229, 240, 241, etc. – ‘(p<0.05)’ – please provide the exact p-value.

24.         After calculating the effect size, add a description in the context of whether the differences were small, medium, or large. Accordingly, revise the description of the results.

25.         I will evaluate the discussion and conclusions after the above corrections are made.

26.         L356 – ‘”.’ Remove the unnecessary quotation mark.

Author Response

Comments 1:  Please correct the reference format. The year should be bold, not the volume. Additionally, the format regarding the number of authors cited, etc., does not meet the journal's standards. Moreover, authors often do not cite the pages they reference, e.g., positions 36, 26, and 16. Therefore, please review the author guidelines and correct the work accordingly.

Response 1: Thank you for your careful review and for pointing out the formatting inconsistencies in the references. We have revised the reference list to ensure that the year is bolded, as per the journal’s standards. Additionally, we have reviewed the author guidelines and corrected the format regarding the number of authors cited.

Comments 2: Approximately 61% of the references are older than 10 years, which is unacceptable. This must be corrected. It is worth noting that in the introduction, positions 1-27 include 15 works older than 10 years. This exceeds half, and authors should refer to the current literature when providing statistical data.

Response 2: Thank you for your observation regarding the age of the references. We have reviewed the cited literature and updated the references to include more recent works, particularly in the introduction.

Comments 3: Identified self-citations: a. Jaemyoung Park – not found. b. Junghyun Kim – position 37. c. Kyeongjin Lee – positions 26 and 36. d. Changho – not found. Note that the described self-citations are acceptable in the work.

Response 3: Thank you for your detailed review and for confirming that the described self-citations are acceptable in the work. Regarding Jaemyoung Park, there was no appropriate section in the manuscript where their work could be cited. Additionally, we have added citations for Junghyun Kim and Changho Song where appropriate. This information can be found on page 1, line 53.

Comments 4: L29-38 – I suggest adding information on stroke epidemiology – e.g., 10.1016/S1474-4422(21)00252-0.

Response 4: Thank you for your valuable suggestion. I have added information on stroke epidemiology to the introduction, incorporating data from the reference you provided (10.1016/S1474-4422(21)00252-0). Specifically, I included details highlighting that stroke remains one of the leading causes of death and disability worldwide, with over 12 million incident cases and 6.55 million deaths reported globally in 2019. Additionally, I noted that despite advancements in treatment, the burden of stroke has increased significantly, particularly in low- and middle-income countries. This addition helps better contextualize our study's importance within the broader global health landscape. This information can be found on page 1, lines 30-33.

“Stroke remains one of the leading causes of death and disability worldwide, with over 12 million incident cases and 6.55 million deaths reported globally in 2019. Despite advancements in treatment, the burden of stroke has increased significantly, particularly in low- and middle-income countries [1].”

Comments 5: L49-50 – ‘Recent technological advancements have spurred various studies aimed at enhancing gait rehabilitation in patients with stroke.’ – this sentence is more suitable for a popular science paper. Please specify what recent technological advancements are being referred to.

Response 5: Thank you for your insightful feedback. I agree that the sentence could be more specific to better suit an academic audience. I have revised the text to specify the technological advancements being referred to. The updated sentence now mentions specific technologies such as robot-assisted therapy, virtual reality rehabilitation, and wearable assistive devices, which have been increasingly utilized in recent studies to enhance gait rehabilitation in stroke patients. These examples provide a clearer understanding of the recent advancements relevant to our study. This information can be found on page 2, lines 52-54.

"Recent technological advancements, such as robot-assisted therapy [12, 13], virtual reality rehabilitation [14, 15], and wearable assistive devices [16-18], have spurred var-ious studies aimed at enhancing gait rehabilitation in patients with stroke."

Comments 6: L4 – ‘One innovative study utilized’ – I suggest avoiding such comments. Whether the study was innovative will be judged by history. Comments like this reduce objectivity and should be avoided throughout the text.

Response 6: Thank you for your suggestion. I agree that such subjective language should be avoided to maintain objectivity. I have revised the sentence to remove the term "innovative" and have ensured that similar comments are avoided throughout the text.

Comments 7:  L64 – ‘electromyographic (EMG)’ – authors refer to surface electromyography; therefore, please change ‘EMG’ to ‘sEMG’. Note for the entire text: the abbreviation ‘EMG’ usually suggests the use of needle electrodes, whereas ‘sEMG’ suggests surface electrodes.

Response 7: Thank you for pointing this out. We agree that 'sEMG' is a more accurate term for our study, as we are indeed referring to surface electromyography. We have revised the text accordingly, replacing 'EMG' with 'sEMG' throughout the manuscript to ensure clarity and accuracy.

Comments 8: Since the authors are based on sEMG, please provide more information about it, how it works, what it analyzes, in which studies it is used, etc. The work 10.1016/j.jelekin.2023.102796 and 10.3390/jcm13051328 might be helpful. For example: 'sEMG works by detecting and analyzing electrical signals that result from physiological changes in the cell membranes of muscle fibers. A key aspect of sEMG is understanding that human tissue, particularly muscle, can generate and conduct electrical impulses fundamental to the muscle contraction process. When a muscle is at rest, it is in a state of electrical equilibrium known as the resting potential. However, during contraction, depolarization of the muscle membrane occurs, meaning there is a flow of ions between the inside and outside of the muscle membrane, generating an electrical signal that is recorded.' – A paraphrase of the above text should be acceptable, citing 10.3390/jcm13051328.

Response 8: Thank you for your insightful suggestion. I agree that providing more detailed information on sEMG would enhance the manuscript. I have incorporated additional details on how sEMG works, what it analyzes, and its applications in research, as per your recommendation. Specifically, I have included the following revised text in the manuscript: (page 2, lines 66-73)

"Surface electromyography (sEMG) is a technique used to measure and analyze the electrical signals of muscle activity through electrodes placed on the skin's surface [24, 25]. By detecting and analyzing these electrical signals, which result from physiological changes in the muscle fibers, sEMG can provide valuable insights into muscle function and its role in movement [25]. A key application of EMG is in the use of electromyographic feedback to encourage specific movements; for instance, a previous study utilized EMG feedback to promote pedaling on the affected side in stroke patients, leading to improved muscle activation and gait on the paralyzed side [23]."

Comments 9:  In work no. 26 cited by the authors, I see that Figure 1 is very similar to Figure 1 in the work I reviewed. Did the authors use the same cohort? Please note this information in the work if so.

Response 9: Thank you for your observation. I understand your concern regarding the similarity between Figure 1 in the current work and Figure 1 in the cited work (reference no. 26). Figure 1 in our manuscript is a CONSORT 2010 flow diagram, which is a standardized format recommended by MDPI. This format is widely used in clinical trial reporting, and as a result, it may appear similar across different studies. However, I want to clarify that although the same type of device was used in both studies, the cohort in this study is not the same as that in the cited work. I will include a note in the manuscript to clarify this and ensure there is no confusion. This information can be found on page 3, lines 103-105.

“This study involved an independent cohort of participants, distinct from those previously studied in related research.”

Comments 10: Add the research hypothesis at the end of the introduction.

Response 10: Thank you for your suggestion. I have added the research hypothesis at the end of the introduction to clearly outline the expected outcomes of the study. This information can be found on page 2, lines 77-84.

“In this study, we hypothesized that the use of electromyographic (EMG) feedback to encourage pedaling on the affected side would lead to significant improvements in muscle strength, balance, gait symmetry, and daily living activities in stroke patients. Conversely, the null hypothesis posited that EMG feedback would not result in significant improvements in these areas.”

Comments 11: L91 – ‘Brunnstrom stages’ – please provide a citation – e.g., 10.3109/02699052.2010.506860.

Response 11: Thank you for pointing this out. I have added a citation to support the reference to "Brunnstrom stages." While I appreciate the suggested citation, I have chosen to cite a source that is more directly related to lower limb motor function, specifically "A Kinematic Data Based Lower Limb Motor Function Evaluation Method for Post-Stroke Rehabilitation," which aligns closely with the context of our study. This information can be found on page 2, line 99.

Comments 12: L98-99 – after this description, I am unable to repeat the analysis in the program. What tests were chosen? What family test, etc.? Please describe it accurately so that anyone can perform the same analyses.

Response 12: Thank you for your valuable feedback. We understand the importance of providing clear and detailed information to ensure that the analysis can be replicated by others.

In determining the sample size, we used the G-power 3.1.9 software.  We chose this test because our study aimed to compare the mean differences between the two groups (CIMCT and GCT). The effect size was set at 0.87, based on the results of the Timed Up and Go (TUG) test from our pilot study. We used a significance level of 0.05 and a power of 0.8, which resulted in a required sample size of 18 participants per group. To account for a potential dropout rate of 10%, we increased the final sample size to 20 participants per group.

 The changes have been made in the revised manuscript on page 3, lines 110-115.

"The sample size for this study was determined based on the results of a pilot study. Before conducting the main experiment, a pilot study was performed with 5 participants in each group, using the same methods as the main study. The effect size was calculated to be 0.87 based on the Timed Up and Go (TUG) test, a dynamic balance variable. The G-power 3.1.9 software was employed to compare the two groups' differences. The significance level was set at 0.05, and the power of the test was 0.8, resulting in a required sample size of 18 participants per group. To account for a potential dropout rate of 10%, the final sample size was increased to 20 participants per group."

Comments 13: While a p-value threshold of 0.05 is standard, please provide information on why the power is set to 0.8. I am not saying it is incorrect, but I need justification.

Response 13: Thank you for your comment. We set the statistical power of the study to 0.8 (or 80%) as this is a widely accepted standard in research design, particularly in randomized controlled trials. A power of 0.8 is commonly chosen because it provides a good balance between the likelihood of detecting a true effect (if one exists) and the risk of committing a Type II error (false negative). This approach is aligned with the recommendations outlined in Zabor et al. (2020), which discuss the rationale for selecting appropriate power levels in clinical research. By setting the power to 0.8, we ensure that there is an 80% chance of correctly rejecting the null hypothesis when the alternative hypothesis is true, which is considered sufficient for most studies and helps maintain a reasonable sample size while achieving adequate sensitivity to detect meaningful effects. I have added this reference to the manuscript to provide further justification for our choice. This information can be found on page 3, line 113.

Reference:

Zabor, E.C.; Kaizer, A.M.; Hobbs, B.P. Randomized controlled trials. Chest 2020, 158, S79-S87.

Comments 14: L99-101 – ‘0.87,’ – was the pilot study published? Please provide a reference. If not, please present the calculations in the response to the reviewer explaining why ES = 0.87 is crucial.

Response 14: Thank you for your inquiry regarding the pilot study. The pilot study that informed the effect size calculation for our main study has not been published, which is why a reference was not provided in the manuscript. The changes have been made in the revised manuscript on page 3, lines 108-112.

The effect size of 0.87 was derived from the results of the Timed Up and Go (TUG) test, a dynamic balance variable measured during our pilot study. In this pilot study, we included 5 participants in each group and applied the same methodology as in the main study. The large effect size observed was crucial for determining the sample size, as it reflected the substantial difference we observed in dynamic balance between the two groups. We used this effect size to ensure that our study would be adequately powered to detect similar differences in the main study.

“The sample size for this study was determined based on the results of a pilot study. Before conducting the main experiment, a pilot study was performed with 5 participants in each group, using the same methods as the main study. The effect size was calculated to be 0.87 based on the Timed Up and Go (TUG) test, a dynamic balance variable.”

Comments 15: L133 – Was the warm-up standardized for everyone? Was the heart rate analyzed during the warm-up? Were oxygen thresholds determined?

Response 15: Thank you for your valuable questions concerning the warm-up procedure (page 5, lines 183-187).

Standardization of Warm-up: Yes, the warm-up was standardized for all participants. Each participant engaged in a consistent warm-up routine, which included light cycling for a duration of 10 minutes at a controlled intensity level. The warm-up was conducted under the supervision of a therapist, who adjusted the routine to suit the participant's condition, ensuring proper execution and adequate preparation for the main exercise without inducing fatigue.

Heart Rate Analysis During Warm-up: Heart rate analysis was not conducted during the warm-up. Instead, the therapist supervised the warm-up closely, adjusting the intensity and duration as needed based on the participant's condition to ensure they were ready for the subsequent exercise without overexertion.

Oxygen Thresholds: Oxygen thresholds were not specifically determined during the warm-up phase. The focus of the warm-up was to ensure participants were adequately prepared for the exercise protocol without pushing them into high-intensity zones. As the primary objective of the study did not require the analysis of oxygen thresholds, this aspect was not included.

Comments 16: L133-137 – why were these specific time intervals chosen?

Response 16: Thank you for your question regarding the selection of specific time intervals. The time intervals were chosen based on recommendations outlined in the statement "Physical Activity and Exercise Recommendations for Stroke Survivors" from the American Heart Association/American Stroke Association. According to these guidelines, many stroke survivors may better tolerate multiple short bouts of moderate-intensity physical exercise (e.g., three 10- or 15-minute exercise sessions) repeated throughout the day, rather than a single long session. This approach, often referred to as interval training or a work-rest approach, was considered optimal for our study to ensure participant safety and maximize the benefits of the intervention. I have included this reference in the manuscript to support our rationale for the chosen time intervals. This information can be found on page 5, lines 185-188.

Comments 17:  At what times was the sEMG study conducted? Was it the same for everyone?

Response 17: Thank you for your question. The training was conducted in alignment with the treatment schedule provided to the inpatients, which included therapy sessions twice a day. Therefore, the training was administered once in the morning or afternoon, depending on the participant's schedule. Although not all participants received the training at the exact same time, the study was conducted with equivalent time intervals between sessions for all participants to ensure consistency. This information can be found on page 3, lines 128-133.

“The training was conducted in alignment with the inpatient treatment schedule, which included two therapy sessions per day. Accordingly, training was administered once in the morning or afternoon, depending on each participant's schedule. Although the exact timing varied between participants, the procedures were executed with consistent time intervals between sessions to ensure uniformity across the study.”

Comments 18: How was the skin cleaned during the sEMG study? Describe in the text.

Response 18: Thank you for your observation. Before applying the sEMG electrodes, the skin was cleaned using alcohol wipes to remove any dirt, oils, or residues that could interfere with signal quality. This step was taken to ensure optimal conductivity and accurate data collection. I have added this information to the manuscript to provide clarity on the preparation process. This information can be found on page 5, lines 171-172.

Comments 19: What electrodes were used during the sEMG study? What was the conductive area? Describe in the text.

Response 19: Thank you for your observation. Before applying the sEMG electrodes, the skin was cleaned using alcohol wipes to remove any dirt, oils, or residues that could interfere with signal quality. This step was taken to ensure optimal conductivity and accurate data collection.

To provide further clarity, we have added the following details to the manuscript. This information has been added to the manuscript on pages 4-5, lines 159-170, to ensure the preparation and application process is clearly described.

“The sEMG electrodes patch (Ag/AgCl surface electrode H2223H, 3M, Maplewood, MN, USA) were placed on the rectus femoris, biceps femoris, tibialis anterior, and gastrocnemius muscles. Each electrode was positioned according to the SENIAM project recommendations as follows:

Rectus Femoris: The electrodes were placed at 50% of the distance along the line from the anterior superior iliac spine to the superior part of the patella.

Biceps Femoris: The electrodes were placed at 50% of the distance along the line between the ischial tuberosity and the lateral epicondyle of the tibia.

Tibialis Anterior: The electrodes were placed at 1/3 of the distance along the line from the tip of the fibula to the tip of the medial malleolus.

Gastrocnemius: The electrodes were placed on the most prominent bulge of the muscle.

Although skin impedance was not directly measured, we minimized it by cleaning the skin with alcohol wipes before electrode application.“

Comments 20: What was the maximum skin impedance accepted? Describe in the text.

Response 20: Thank you for your question regarding skin impedance and its potential impact on sEMG signal normalization. The clinical trial utilized the MyoWare device, which has a high input impedance of 110 GΩ, ensuring that the sEMG signals were accurately captured without significant interference from the device itself. While we did not specifically measure skin impedance before applying the electrodes, we ensured that proper skin preparation (cleaning with alcohol wipes) was performed to minimize skin impedance and maintain high-quality signal acquisition. Although skin impedance can affect signal quality, it does not directly influence the normalization process itself. Instead, it is important to ensure that the sEMG signals are reliable and free from excessive noise or distortion. Given the careful measures we implemented to ensure good electrode contact and signal quality, we believe that skin impedance did not significantly impact the normalization of the sEMG signals in this study. This information can be found on pages 4-5, lines 157-159, 171-172.

Comments 21: Was an interference test conducted during the sEMG study? Describe in the text.

Response 21: Thank you for your question regarding interference testing during the sEMG-triggered CIMCT study. To minimize potential interference, we ensured that the sEMG data was collected in a space free from other electronic equipment that could introduce noise. Additionally, to reduce interference caused by the sEMG cables touching each other, we used mesh coverings on the cables. These precautions helped us maintain high-quality, interference-free signal acquisition throughout the study. This information can be found on page 4, line 153-156.
“To minimize interference during the sEMG-triggered CIMCT study, we conducted the data collection in a space free from other electronic equipment. Additionally, to reduce interference caused by the sEMG cables touching each other, mesh coverings were applied to the cables.”

Comments 22: L213 – there is a slight mix-up here; MDC is not effect size. MDC refers to the smallest difference considered statistically significant in the context of measurement. It is a measure of the sensitivity of a measurement tool or test procedure. Effect size refers to the magnitude of the difference or relationship being studied. It is a measure of the strength of a given effect and can be expressed in various ways depending on the type of data and statistical tests. Therefore, the authors should refer to effect size in the statistical description and results for each p-value. ‘Effect size helps readers understand the magnitude of differences found, whereas statistical significance examines whether the findings are likely to be due to chance.’ - 10.4300/JGME-D-12-00156.1. For a t-test, this will be Cohen’s d: small 0.20, medium 0.50, large 0.80, and for the chi-square test, Cramér’s V and phi (φ) depending on the degrees of freedom (df-1): small 0.1, medium 0.3, large 0.5. Work 10.3390/jpm14060655, tables 4 and 5 might be useful. I believe this reference may be helpful.

Response 22: Thank you for your detailed and insightful feedback. We appreciate the clarification regarding the distinction between Minimal Detectable Change (MDC) and effect size. In response to your comment, we have carefully reviewed and revised the manuscript to ensure accurate usage of these terms (Tables 2,5, and 6) (pages 8, 10, 11 lines 289-291, 302-305, 312-313).

For the tables that show significant differences Tables, we have added the corresponding effect sizes to better convey the magnitude of these differences. Specifically, we have used Cohen’s d for t-tests, with interpretations aligned with commonly accepted thresholds: small (0.20), medium (0.50), and large (0.80) effect sizes, as you suggested.

Comments 23: L229, 240, 241, etc. – ‘(p<0.05)’ – please provide the exact p-value.

Response 23: Thank you for your suggestion. We agree that providing the exact p-values enhances the clarity and precision of the statistical reporting (Tables 2, 3, 4, 5, and 6).

In response to your comment, we have updated the tables in the manuscript to include the exact p-values to clearly indicate the statistical significance of the findings.

Comments 24: After calculating the effect size, add a description in the context of whether the differences were small, medium, or large. Accordingly, revise the description of the results.

Response 24: Thank you for your suggestion. We recognize the importance of providing context for the effect sizes to enhance the interpretation of our findings.

In response to your comment, we have updated the results section to include a discussion of the effect sizes, highlighting the interpretation of small, medium, and large values according to the commonly accepted thresholds.This information can be found on pages 8, 10, 11 lines 289-291, 302-305, 312-313.

“The strength of the knee flexor, extensor, dorsiflexor, and plantar flexor muscles on the affected side significantly increased after training in the CIMCT group (p<0.05), with Cohen's d values indicating a large effect size (d > 0.80).”

”For the TUG, BBS, and FRT, the CIMCT group demonstrated a significant reduction after training (p<0.05), and the GCT group also displayed a significant decrease after training (p<0.05). Additionally, Cohen's d values for BBS and FRT indicated a large effect size (d > 0.80).”

”For the MBI, both the CIMCT and GCT groups demonstrated a significant increase after training (p<0.05), with Cohen's d values indicating a large effect size (d > 0.80).”

Comments 25:  I will evaluate the discussion and conclusions after the above corrections are made.

Response 25: Thank you for your patience and consideration. We have made the necessary corrections as suggested and look forward to your evaluation of the discussion and conclusions.

Comments 26: L356 – ‘”.’ Remove the unnecessary quotation mark.

Response 26: Thank you for pointing that out. The unnecessary quotation mark has been removed from the text to correct the formatting error.

Round 2

Reviewer 1 Report

Comments and Suggestions for Authors

I acknowledge the diligent effort the authors have made in addressing the comments and suggestions provided on the previous version. The manuscript demonstrated significant improvement in terms of clarity and overall presentation. There are a few additional comments that could further enhance the manuscript.

·       Abstract: I would reiterate one of the previous comments. A clear delineation of the primary and secondary outcome measures, including the specific instrument employed for each is essential to facilitate comprehensibility. In the authors' response to this comment, they did not specify the primary and secondary outcomes (i.e. measured variables) and the instruments used for the assessment of each of them. They only mentioned the intervention allocation.

·       Abstract: It would be very interesting if authors could include the effect sizes (a clinical measure) besides the p-value (a statistical measure).

·       The authors noted in their response that they re-analyzed using repeated-measure ANOVA and the results were consistent with those of the t-test. However, for the sake of accuracy and credibility of the data, I suggest replacing the t-test results with those obtained using repeated measures ANOVA.

Author Response

Comments 1: Abstract: I would reiterate one of the previous comments. A clear delineation of the primary and secondary outcome measures, including the specific instrument employed for each is essential to facilitate comprehensibility. In the authors' response to this comment, they did not specify the primary and secondary outcomes (i.e. measured variables) and the instruments used for the assessment of each of them. They only mentioned the intervention allocation.

Response 1: Thank you for your valuable feedback. I agree with your comment regarding the need to clearly specify the primary and secondary outcomes along with the instruments used. Therefore, I have revised the abstract to include a clear delineation of the primary and secondary outcome measures and the specific tools employed. The change can be found on page 1, lines 19-29.

“The primary outcome measures showed a significant increase in muscle strength on the affected side in the CIMCT group, as assessed by a manual muscle tester (p<0.05), with a large effect size (d = 1.86), while no meaningful improvement was observed in the GCT group. Both groups demonstrated significant improvements in dynamic balance, as measured by the Timed Up and Go (TUG) test (p<0.05), with the CIMCT group showing superior results compared to the GCT group, reflected by a large effect size (d = 0.96). The secondary outcome measures revealed significant improvements in activities of daily living in both groups, as assessed by the Modified Barthel Index (MBI), with the CIMCT group achieving the substantial improvement (p<0.05), accompanied by a large effect size (d = 0.87). This study concludes that sEMG-triggered CIMCT effectively improved muscle strength, postural balance and activities of daily living in patients with chronic stroke.”

Comments 2: Abstract: It would be very interesting if authors could include the effect sizes (a clinical measure) besides the p-value (a statistical measure).

Response 2: I appreciate your suggestion to include effect sizes as a clinical measure alongside p-values. I have incorporated effect sizes (Cohen’s d) for the key outcome measures in the abstract. The change can be found on page 1, lines 19-29.

 “The primary outcome measures showed a significant increase in muscle strength on the affected side in the CIMCT group, as assessed by a manual muscle tester (p<0.05), with a large effect size (d = 1.86), while no meaningful improvement was observed in the GCT group. Both groups demonstrated significant improvements in dynamic balance, as measured by the Timed Up and Go (TUG) test (p<0.05), with the CIMCT group showing superior results compared to the GCT group, reflected by a large effect size (d = 0.96). The secondary outcome measures revealed significant improvements in activities of daily living in both groups, as assessed by the Modified Barthel Index (MBI), with the CIMCT group achieving the substantial improvement (p<0.05), accompanied by a large effect size (d = 0.87). This study concludes that sEMG-triggered CIMCT effectively improved muscle strength, postural balance and activities of daily living in patients with chronic stroke.”

Comments 3: The authors noted in their response that they re-analyzed using repeated-measure ANOVA and the results were consistent with those of the t-test. However, for the sake of accuracy and credibility of the data, I suggest replacing the t-test results with those obtained using repeated measures ANOVA.

Response 3: Thank you for your suggestion. I agree that replacing the t-test results with those from the repeated measures ANOVA enhances the accuracy and credibility of the data. We have now replaced the t-test results with those obtained using repeated measures ANOVA in the manuscript. The change can be found on pages 7, 8, 10-11, tables 2, 5, 6, lines 275-279.

"The changes in the dependent variables before and after the intervention were analyzed using paired t-tests. For instances where significant differences were observed, re-peated measures ANOVA was employed to compare the effects between the two groups and analyze the interaction effects between group and time. "

Reviewer 2 Report

Comments and Suggestions for Authors

Thank you for your reply. I accept the answers and improvements. I have no further comments. 

Author Response

Thank you very much for your constructive feedback. Your detailed review has been instrumental in improving the quality and precision of our manuscript. We are grateful for your suggestions, which have enabled us to enhance the scientific rigor and reliability of our findings.